# Modelling Lactation Curves for Dairy Sheep in a New Zealand Flock

**DOI:** 10.3390/ani13030349

**Published:** 2023-01-19

**Authors:** Ana Carolina Marshall, Nicolas Lopez-Villalobos, Simon M. Loveday, Ashling Ellis, Warren McNabb

**Affiliations:** 1Department of Animal Science, School of Agriculture and Environment, Massey University, Private Bag 11 222, Palmerston North 4442, New Zealand; 2The Riddet Institute, Massey University, Private Bag 11 222, Palmerston North 4442, New Zealand; 3Singapore Institute of Food and Biotechnology Innovation (SIFBI), Agency for Science, Technology and Research (A*STAR), Singapore 138673, Singapore; 4Smart Foods Innovation Centre of Excellence, AgResearch Ltd., Private Bag 11 008, Palmerston North 4442, New Zealand

**Keywords:** dairy sheep, animal model, lactation curve, milk production, lactation persistency

## Abstract

**Simple Summary:**

The New Zealand dairy sheep industry is relatively new, and more studies are needed to support New Zealand dairy sheep farms for more efficient production. The objective of the present study was to model the lactation curves of dairy sheep from a New Zealand commercial flock, providing information on the lactation yields of milk, fat, protein, and lactose. The factors affecting the lactation yields and lactation persistency of dairy sheep were described. The low production per ewe and the large variation between animals reported in the present study indicate that there is room for improvement in lactation yields and in lactation persistency in dairy sheep in New Zealand, and this could be achieved through the improvement of farming systems and through the building of a national genetic improvement programme targeted at dairy sheep.

**Abstract:**

Lactation curves were modelled for dairy sheep in a New Zealand flock, providing information on the lactation yields of milk, fat, protein, and lactose, corrected for 130 days of milking. From 169 ewes, a total of 622 test-day records were obtained during the milk production season of 2021–2022 (from October to January). The flock produced an average of 86.1 kg of milk, 5.1 kg of fat, 4.5 kg of protein, and 4.1 kg of lactose, and moderate to large coefficients of variation were observed (27–31%) for these traits. The lactation persistency of milk, fat, protein, and lactose yields ranged from 52.3 to 72.7%. Analyses of variance for total yield and persistency were performed with an animal model that included the fixed effects of age (parity number), litter size, coat colour, and milking frequency (days in twice-a-day milking) and random residuals. Age and milking frequency were the only factors that significantly affected the yields of milk, fat, protein, and lactose. Age significantly affected the lactation persistency of milk and lactose yields, whereas litter size affected the persistency of protein, and milking frequency affected the persistency of fat. This study on this single flock provides valuable experience for a larger-scale animal breeding programme in New Zealand.

## 1. Introduction

The dairy industry in New Zealand is characterised by being pasture-based, with no housing and low supplementation [1]. The dairy sheep industry is expanding, and it is expected to reach NZ$750 million in annual export receipts by 2035 [2]. Milking sheep is potentially gentler to the environment than traditional dairy cow farming [3]. In addition, dairy sheep milk contains more solids than cow milk and provides several health benefits to consumers due to its composition [4]. 

A small number of studies have described the milk production of dairy sheep in some New Zealand flocks [5,6,7,8], reporting production levels that are lower than dairy sheep in well-advanced production systems in Europe. This has been suggested to be due to the small pool of genetics that were initially imported from Europe, where animals were originally selected for intensive farming systems [1].

Test-day records can be used to predict the total amounts of milk, fat, protein, and lactose produced by each animal throughout lactation. Test-day records are taken at fixed dates, but individual animals will be at different days in milk, because they started producing milk at different dates. For example, a single test-day might be on day 20 of the lactation for an animal that gave birth 20 days ago, and simultaneously, it is day 45 for an animal that gave birth 45 days ago. These animals are not in the same stage of lactation and are not comparable, given that the stage of lactation significantly affects milk yield and composition [9]. One way to overcome this problem and estimate total yields for individual animals is through the modelling of the lactation curve using random regression with Legendre polynomials [10,11,12]. This technique models the covariance between repeated records taken on the same animal over time and allows the prediction of variances and covariances for timepoints along the trajectory, even though few observations are made, but using information from all other measurements [13,14]. This approach is considered ideally suited for the analysis of longitudinal data in animal breeding [10,15].

Lactation curves can be used to calculate lactation persistency, which indicates the ability of animals to maintain a reasonably constant milk yield after peak production. Several methods for the calculation of lactation persistency have been proposed, but there is still no standard method [16,17,18].

The total yield obtained for each animal can be used for the analysis of variance. The explanatory variables or factors that are known to affect milk performance are included as fixed (e.g., breed and parity number) or random (e.g., herd and animal) effects. Several factors that may affect milk production and the shape of the lactation curve include genetics (such as the breed and individual genotype), physiological factors (such as age or parity, liveweight, health, and offspring characteristics such as litter size), and farm management (such as milking frequency and nutrition) [19].

The objective of this study was to model the lactation curves of dairy sheep in a New Zealand commercial flock and identify the factors affecting the shape of the lactation curve and the total yields and persistency of milk, fat, protein, and lactose yields. 

## 2. Materials and Methods

Data were collected from 169 Dairymeade ewes at Kingsmeade Farm, Masterton, New Zealand. Animal ethics approval was obtained for this study (Massey University Animal Ethics Committee-Protocol 21/45). The breed was established in 1996, initially using Coopworth and Border Leicester dams and semen from European East Friesian sires. The resulting progenies were subsequently mated with only East Friesian sires [20]. The farm has been using the self-replacement of ewes and rams over the past 12 years. In Dairymeade ewes, only two colour variants are visually distinguished, white and black.

The farm has 11 hectares of land, is located on flat land, and operates an extensive seasonal pasture-based system; the ewes have limited access to supplementary feed during milking. Rams are left to mate with mixed-age ewes (over two years old) in mid-March, with ewe hoggets in mid-April and the lambing season starting in mid-August. The farm has an exclusive suckling period, so milking for artisan cheese production starts after the lambs are fully weaned when they reach 13.5 kg liveweight (at the discretion of the farmer). In the 2021–2022 season, the average suckling period was 57 days. 

Masterton has a mild temperate climate that resembles a Mediterranean climate; warm dry settled weather predominates in summer, and frosts may occur in winter. In the 2021–2022 season, the maximum temperatures ranged from 24.7 °C to 30.2 °C, with January being the hottest month. Minimum temperatures ranged from −0.2 °C to 7.4 °C from October 2021 to January 2022. The total monthly rainfall was lowest in January 2022 (13.4 mm) and highest in December 2022 (178.8 mm) (Meteorological Service of New Zealand Ltd., Wellington, NZ, 2022) [21].

### 2.1. Test-Day Records

Collections of test-day records started after the lambs were fully weaned. A total of 622 test-day records were gathered from 169 ewes between October 2021 and the end of January 2022 to obtain 2 to 4 milk tests from each ewe during the milk production season. The flock was dried off in mid-February. 

Milk yields of individual animals were manually recorded from the volume taken from test buckets in the afternoon (at 2:30 p.m.) and on the following morning (at 6 am) in October, when milking was happening twice a day (TAD). On the 1st of November 2021, the milking frequency was shifted to once a day (OAD) at the discretion of the farmer, happening in the afternoons only (at 2:30 p.m.). Reducing the milking frequency close to summer is a common practice on this farm to align with pasture availability. On each test day, a representative milk sample was taken from each animal after measuring milk yield. 

Lambing dates ranged from the 26th of July to the 6th of November 2021, and the median lambing date was the 20th of August 2021. The deviation from the median lambing date of the flock was calculated for each ewe (ewe lambing date–median lambing date of the flock). Due to the shift to OAD milking in November, late-lambing ewes were not milked TAD in early lactation. To adjust for the milking frequency, days in TAD milking were calculated for each ewe (1st of November–weaning date). Information on litter sizes and the ages of animals were supplied to the study. 

### 2.2. Milk Composition

Milk samples were analysed by Milk Test NZ Ltd. (Hamilton, NZ) using a Combi FOSS instrument (Foss Analytics, Hillerod, DK). The composition analysis included fat (%), protein (%), lactose (%) [22], and somatic cell count (cells/mL) [23]. The yield of milk solid components (fat, protein, and lactose) on each test day was calculated by multiplying the milk volume by the concentration of milk solid components.

### 2.3. Pasture Analyses

The flock had access to fresh white clover (*Trifolium repens*)/lucerne (*Medicago sativa*) pasture, and whole-grain maize and wheat were given as supplements during milking (approximately 600 g of supplement feed per milking). Samples of fresh pasture were taken for analysis on milk-sampling days by hand-plucking throughout the paddock where the animals were grazing the day before. These samples were freeze-dried and ground (Wiley mill) and analysed by the Nutrition Laboratory at Massey University (Palmerston North, NZ) using a near-infrared reflectance spectroscopy technique [24] using an MPA Analyser (Bruker Corporation, Billerica, Massachusetts, USA) to evaluate metabolisable energy (ME), crude protein (CP), neutral detergent fibre (NDF) content, and organic matter digestibility (OMD).

### 2.4. Modelling Lactation Curve in Dairy Sheep

Records of milk, fat, protein, and lactose yields of all animals were plotted against time. Time was defined as d = days in milk–35, as most records were made 35 days after the lambing date due to the exclusive suckling period. The shape of the curve of the plotted data was then examined. 

Legendre polynomials were chosen to standardise values to the interval [−1, …., 1], and the coefficients were then calculated using the Rodrigues formula [25]:P0(t)=1,
P1(t)=x,
$$P2(t)=\frac{1}{2}(3x^{2}-1),$$
$$P3(t)=\frac{1}{2}(5x^{3}-3x),$$
$$P4(t)=\frac{1}{8}(35x^{4}-30x^{2}+3),$$
$$P5(t)=\frac{1}{8}(63x^{5}-70x^{3}+15x)$$
where x = −1 + 2, and $\frac{\left(t\;-\;\mathrm{tmin}\right)\;}{\left(\mathrm{tmax}\;-\;\mathrm{tmin}\right)}$, with tmin = 1 and tmax = 130.

Day 1 corresponded to day 35 of lactation. 

The random regression model was represented as follows:$$y_{\mathrm{ti}}=(\beta_{0}P_{0}+\beta_{1}P_{1}+\beta_{2}P_{2}+\cdots +\beta_{n}P_{n})+(\alpha_{0i}P_{0}+\alpha_{1i}P_{1}+\alpha_{2i}P_{2}+\cdots +\alpha_{\mathrm{ni}}P_{n})+\mathrm{eti},$$
where β values are the regression coefficients of the lactation curve of the population, α values are random regression coefficients describing the lactation curve for animal i, n is the maximum polynomial order, and e_ti_ is the random residual for animal i at time t. The estimates of β and α were obtained using the MIXED procedure of SAS version 9.4 (SAS Institute Inc., Cary, NC, USA) [26] with the COVTEST option for covariance parameter estimates. Polynomials of orders 2, 3, 4, and 5 were tested. Based on the Akaike (AIC) and Bayesian (BIC) information criteria (smallest is the best), an orthogonal polynomial of order 4 was considered the best fit for modelling daily milk and lactose yields. An orthogonal polynomial of order 5 was considered the best fit for modelling fat and protein daily yields. The best covariance structure of random residuals was a diagonal-constant matrix (Toeplitz) for the modelling of repeated records on the same animal, also based on AIC and BIC values [27].

The somatic cell score (SCS) was calculated for each test-day record as SCS = log_2_(SCC). It was not possible to model the lactation curve for SCS, as the records did not follow any pattern throughout lactation. Thus, the average SCS was calculated for each lactation.

After choosing the best order of fit and the covariance structure for the models and computing the regressor coefficients (α_0_ to α_4_ or α_0_ to α_5_, depending on the trait) of each ewe, the daily milk yield was predicted from 35 to 164 days in milk (or from t = 1 to t = 130) using Legendre polynomial models of order 4 for milk and lactose yields and order 5 for fat and protein yields. Then, the predicted yields on each day of the lactation were summed to obtain an estimated total milk yield produced by each ewe for 130 days (from day 35 to day 164 of lactation). The measure of lactation persistency was defined as the estimated yield produced from day 101 to 164 divided by the estimated yield produced from day 35 to day 100 and expressed as a percentage. The higher this ratio, the higher the persistency. More persistent lactation will have a flatter curve, with the persistency proportion approaching one.

### 2.5. Measures of Goodness of Fit

The actual (A) and predicted (P) values for milk, fat, protein, and lactose yields were compared using linear regression of the actual on predicted values using PROG GLM in SAS version 9.4 software (SAS Institute Inc., Cary, NC, USA) [26] to obtain the slope and the mean square error (MSE). The relative prediction error (RPE) was calculated as the square root of MSE divided by the mean of the actual values ($\mu_{A})$, multiplied by 100, as per the following equation [28]:$$\mathrm{RPE}=\frac{\sqrt{\mathrm{MSE}}\;}{\mu_{A}}\times 100.$$

The correlation coefficient indicates the closeness of actual and predicted values and was obtained using PROC CORR in SAS version 9.4 software (SAS Institute Inc., Cary, NC, USA) [26]. To evaluate the agreement between paired readings, the concordance correlation coefficient between predicted and actual values was calculated as follows [29]:$$\rho_{\mathrm{ccc}}=\frac{2\sigma_{\mathrm{PA}}}{\sigma_{P}^{2}+\sigma_{A}^{2}+{\left(\mu_{P}-\mu_{A}\right)}^{2}},$$
where $\sigma_{\mathrm{PA}}\;$ is the covariance, $\sigma_{P}^{2}$ and $\sigma_{A}^{2}$ are the variances, µP and µA are the means, and P and A refer to predicted and actual values.

### 2.6. Statistical Analysis

All statistical analyses were performed using the statistical package SAS version 9.4 (SAS Institute Inc., Cary, NC, USA) [26]. Descriptive statistics (mean, standard deviation, minimum and maximum values, and coefficient of variation) for total yields were obtained with the MEAN procedure. Analysis of variances for the estimated total yields and regression coefficients were performed using the MIXED procedure with a linear model that included the fixed effects of ewe coat colour (i = categorical variable with two levels: black or white) as an indicator of animal variety within the breed, litter size (categorical variable with two levels: j = 1 lamb or 2 lambs and greater), and ewe age class or parity number (categorical variable with four levels: k = 1, 2, 3, and 4 years and older), with days in TAD milking as a covariate ($\beta_{1})$. The covariate days in TAD was used in the model instead of the deviation from the median lambing date, as these traits were strongly correlated (−0.6). The model is represented as follows:$${\;y}_{\mathrm{ijkl}}=\mu +{\;\mathrm{coat}\;\mathrm{colour}}_{i}+{\;\mathrm{litter}\;\mathrm{size}}_{j}+{\;\mathrm{age}}_{k}+\beta_{1}{\mathrm{days}\;\mathrm{in}\;\mathrm{TAD}}_{\mathrm{ijkl}}+{\;e}_{\mathrm{ijkl}}$$
where ${\;y}_{\mathrm{ijkl}}$ represents the dependent variables, which include the estimated total yields of milk, fat, protein, and lactose, the regression coefficients of individual animals, and lactation persistency. Least-squares means for each class of fixed effects and standard errors were obtained and used for mean comparisons using Fisher’s least significant difference test.

## 3. Results

### 3.1. Pasture Quality

The level of metabolisable energy (ME) in the pasture dry matter was high (12 MJ/kg) in early October and dropped throughout the season to 8.2 MJ/kg in late January (Table 1). The level of crude protein in the pasture dry matter also dropped from 26.7 to 13.5%DM throughout the season, and the levels of non-detergent fibre increased from 36.5 to 60.6%DM. Consequently, organic matter digestibility decreased from >84 to only 57.6%.

### 3.2. Average Production

The mean lactation length of the flock in the present study was 165 days, and the mean milking length was 108 days. The flock produced an average of 86.1 kg of total milk yield per ewe (0.7 kg per ewe/day) and an average of 5.1 kg of total fat, 4.5 kg of total protein, and 4.1 kg of total lactose yield per ewe for 130 days of milking (Table 2).

### 3.3. Model Adequacy

The measures of goodness of fit are presented in Table 3. The intercepts of the regression lines of the actual on predicted values were negative, but not departing significantly from zero, and the slopes were all greater than 1.0, with relative predicted error close to 10%. The correlation and concordance correlation coefficients were close to one.

### 3.4. Factors Affecting the Lactation Curves

The F- and *p*-values for fixed effects are presented in Table 4. Least-squares means for lactation yields and for lactation persistency for the different class effects are presented in Table 5 and Table 6. Least-squares means for the regression coefficients are provided in Appendix A. The effects of age (parity number) and days in TAD milking were significant on all total yields (Table 4). The factor that had the greatest effect on the total yields was days in TAD milking (largest F-values).

One-year-old ewes produced significantly less milk, fat, protein, and lactose than older ewes (Table 5). Three-year-old ewes produced the highest yields of milk, protein, and lactose, whereas four-year-old ewes produced the highest yield of fat. Ewes that lambed late in relation to the median lambing date of the flock missed TAD milking in early lactation and produced significantly less milk, fat, protein, and lactose than early-lambing ewes that were milked TAD in early lactation, with the largest F-value for the effect of days in TAD on total fat yield (27.47).

Ewe age affected the lactation persistency of milk and lactose yields, litter size affected the lactation persistency of protein yield, and milking frequency strongly affected the lactation persistency of fat yield with a large F-value (10.96). Late-lambing ewes that were not milked TAD in early lactation had a significantly higher lactation persistency of fat than early-lambing ewes (Table 6). Coat colour had no significant effect on any trait. None of the factors significantly affected the somatic cell score.

### 3.5. Lactation Curves

Lactation curves for daily yields of milk, fat, protein, and lactose for different ewe ages are presented in Figure 1. Overall, milk, fat, protein, and lactose yields declined over the course of lactation (from 35 to 164 days in milk), and no initial peak was observed. This descending trend is not only a physiological occurrence, but it is also likely to be a result of reduced pasture quantity and quality in this seasonal pasture-based system. An atypical increase towards the end of the season (between 100 and 150 days in milk) was observed in the lactation curves for different ewe ages, mainly for fat and protein yields (Figure 1B,C). Lactation curves for one-year-old ewes are visually distinct from the lactation curves of two-, three-, and four-year and older ewes (Figure 1). Overall, milk yield increased with age (parity), with 3-year-old ewes producing the highest yields, which then descended afterwards, producing more than 4-year-old ewes. 

## 4. Discussion

### 4.1. Flock Performance

There are scarce scientific publications on the milk production of dairy sheep in New Zealand to enable comparisons, but the low daily production per ewe in this flock is noticeable. The low total yield achieved per ewe can be attributed to the long exclusive suckling period, which lasted, on average, 57 days. In addition, the rapid decline in milk production was largely influenced by reduced pasture quality in this seasonal pasture-based system. The composition percentage was within the range reported previously for East Friesians [9,30]. However, due to the lower total milk yield, this flock also produced low total yields of fat, protein, and lactose compared to other studies in New Zealand and overseas [6,8,30,31].

Although higher or similar yields have been reported for other New Zealand flocks, those studies had a short lamb suckling period [6,32,33] or were in a mixed regime of suckling and milk collection [8]. Gosling et al. [32] reported 116 L over 147 days and higher solids for New Zealand Dorset ewes that lambed in spring, with lambs removed at four days of age. The present flock produced a similar total milk yield to those reported for Poll Dorset ewes (86.8 L/ewe) and East Friesian crossed ewes (113.1 L/ewe) milked for 102 days, but their lambs were artificially reared [33]. McMillan et al. [6] reported a much higher milk yield (310 L of milk produced over 147 days) in East Friesian crossed ewes and hoggets in a New Zealand pasture-based system, with no suckling period. More recently, Scholtens et al. [8] reported 234 kg for 126 days of lactation in New Zealand East Friesian crossed ewes.

The low daily production per ewe achieved in this extensive farming system can also be attributed to low forage availability in the summer caused by hot and dry weather conditions. The use of a long exclusive suckling period compromised the total milk yield per ewe as the initial peak of milk production was missed. It has been suggested that a full suckling period during the first 30 or 60 days of lactation can lower yields by 20–25% or 45–50%, respectively [6]. It is important to mention, however, that multiple factors are involved in a farmer’s choice of a weaning system, and these are based not only on the total yield produced but also on the cost, work, stress, and net benefit.

In addition, the breed used in the present study is mainly founded from East Friesians, which was the first dairy breed of sheep to be imported to New Zealand in the 1990s and provided an opportunity for increasing milk yield in sheep dairying [33,34]. Selection based upon temperament, health, and lactation length has been in place at this farm since 1996 and contributed to the establishment and improvement of the Dairymeade breed. However, East Friesians are known for being a high-yielding dairy breed originally selected for intensively managed systems and may have limited production on extensive systems. Under hot or dry conditions, East Friesian ewes have been noted to produce low to moderate yields, and poor adaptability to the Mediterranean climate and semi-arid conditions has been reported [35,36]. East Friesians managed in intensive or mixed indoor–outdoor systems can express their full genetic potential for milk production, and high milk yields of over 400 kg per lactation have been reported [31]. 

The persistency of fat yield was the highest, followed by protein, milk, and lactose yields, agreeing with the findings of Jonas et al. [37]. Better conditions for pasture growth in the middle towards the end of lactation (December) are likely to have affected the persistency of fat and protein yields obtained in the present study, as these traits are largely influenced by the level of feeding [38]. Although there is no single reference method for the calculation of lactation persistency [17,18,39], more persistent lactation will have a flatter curve, with the persistency proportion approaching 100%. The milk yield persistency of this flock was lower and fat and protein yield persistency was higher than that reported by Scatà et al. [40] using a similar method for calculation. 

### 4.2. Model Adequacy and Lactation Curves

All measures of goodness of fit presented in Table 3 indicate that random regression with Legendre polynomials is an adequate technique to model the lactation curves of dairy sheep in this commercial flock. The estimates of the intercepts were close to zero, and the slopes were all slightly greater than 1.0, meaning that the models tended to over-predict at low actual values and under-predict at high actual values, creating RPEs greater than 10%. According to Fuentes-Pila [41], an RPE < 10% is considered satisfactory, and between 10 and 20% is relatively good for prediction models. However, Lin’s concordance correlations [29] close to one indicate that the actual and predicted values were in high agreement with low biases in the mean and regression line of the actual on predicted values.

The typical lactation curve is represented by a rapid increase to a peak in the first few weeks of lactation before gradually declining until the end of lactation. This pattern has been reported in various breeds of sheep [42,43]. The farm used in this study has an exclusive suckling period when there is no milk collection, meaning that few milk yield records were taken in the very early stages of lactation, and this initial peak was not observed for most of the individual lactation curves. 

Overall, milk yield, fat, protein, and lactose yields declined as lactation progressed from day 35 to 164. An atypical small increase and stabilisation at the end of lactation were observed in the lactation curves of fat and protein yields of ewes of 2 years or older (Figure 1B,C), being more obvious for fat yield, as the fat content is the most variable component in milk, and changes are more pronounced with the feeding level [38].

Others have reported “atypical” lactation curves defined as continuously decreased milk production without a lactation peak, even when sampling was performed in the first week of lactation [44]. In pasture-based systems, animals that have low forage availability and low supplementation are not able to fully express their productive potential [45], and the peak may not be observed. The peak is also not observed in less selected animals [46]. In grazing systems, weather affects not only pasture availability but also comfort and stress in animals, therefore also playing an important role in the shape of lactation curves [47].

The lactation curves obtained in this study had a moderate persistency, and a rapid rate of decline in milk production was observed. In theory, an ideal lactation curve for greater milk yield production would have a high peak and a moderately flat trend afterwards. However, correlations between peak yield and persistency have been reported to be negative in cows [48] and in sheep [49]. Additionally, peak yield has been reported to be more correlated with a high lactation yield than with lactation persistency [50].

On the other hand, a very high peak yield has been associated with a highly negative energy balance and metabolic stress in early lactation, and sheep tend to reduce their milk production more markedly than cows when in a negative energy balance [51]. In addition, flatter curves have been related to better animal health and a reduction in feeding costs [17,40,52]. Therefore, the peak and persistency should be carefully considered when selecting dairy animals for milk production. Furthermore, feed management decisions during lactation are likely to mask the real persistency [18].

### 4.3. Animal Factors

Milk yield increased with age (parity), but 3-year-old ewes produced more than 4-year-old ewes. The parity number has been widely reported to significantly affect milk production [49,50,53,54]. The mammary glands of primiparous ewes are still not fully developed, and therefore, they have a less pronounced peak yield and are flatter in shape [55]. 

One-year-old ewes produced 20.9, 0.9, 0.9, and 1.1 kg less milk, fat, protein, and lactose yields, respectively, than 3-year-old ewes. Three-year-old ewes produced more than 4-year-old ewes. However, one-year-old ewes had higher lactation persistency than older ewes, agreeing with other published results [50,56]. One-year-old ewes’ lactation persistency of milk and lactose was 7.8% and 7.1% higher than that of two-year-old ewes, and the effect was significant.

Notably, from a simple analysis of Pearson correlations between effect variables, age (parity) was positively correlated with litter size (0.27) and with days in TAD milking (0.34). This means that older ewes tended to give birth to multiple lambs and to lamb earlier than primiparous ewes. Litter size is reportedly known to be smaller for primiparous ewes compared to multiparous ewes [50,57]. The mating management of the farm, where ewe hoggets are mated later than mature ewes, is likely to be the main determinant of the lambing date in this flock. However, it is also known that mature ewes tend to get pregnant faster than young ewes in the mating season and therefore also lamb earlier in the lambing season. Young ewes can also display shorter and less intense oestrus periods [58] and need to be served by the ram on at least three occasions [59].

There was no difference (*p* > 0.05) in milk yield between ewes with twin lambs and ewes with a single lamb. However, there was a trend for ewes that lambed twins to produce about 2.8 kg more milk than equivalent ewes with a single lamb. Other studies have shown that litter size significantly affects milk production, in that ewes with twins or triplets yield more milk than single-lambing ewes [44,54]. In the current study, ewes with twins produced the same fat yield and slightly higher protein and lactose yields than ewes with a single lamb, though this was also not significant. Others have shown the contents of fat and protein to significantly vary with the effect of litter size [60].

Ewes that carry twins are expected to produce more milk because of their higher secretion of placental lactogen due to a higher placental mass, stimulating the greater development of the mammary gland [61]. Additionally, the higher stimulus of the mammary gland during suckling by lambs stimulates the higher production of milk [62]. However, it has been suggested that this is observed in the first stage of lactation, mainly until the peak yield, showing no significant differences after the 10th–17th week of lactation [44,63]. In the current study, few records were made in the very early stage of lactation due to the suckling period, and the lactation peak was not observed; therefore, differences in milk production due to litter size were not pronounced. 

Interestingly, litter size did not significantly affect the lactation persistency of milk, fat, or lactose yields but affected the lactation persistency of protein yield, even though litter size had no significant effect on protein yield. Ewes that had twin lambs had a sharper decline in protein yield during lactation compared with ewes that had a single lamb. Previous studies have shown a small litter size to be correlated with a lower total milk yield and with flatter lactation curves [50,64]. 

In this study, coat colour did not significantly affect milk production. However, there was a trend for white ewes to produce 1.2 kg more milk yield than ewes with black colouring (Table 5). Only 24% of the flock were black-coated due to the white colour being dominant over black/brown coats in various breeds of sheep [65]. Other studies have shown the colour/variety to be associated with production traits in other sheep breeds [46,63,66], indicating the effect of the genotype on production. In the study by Peralta-Lailson et al. [46], the variety significantly affected the milk yield of Creole sheep in Mexico, with brown ewes producing more than white and black ewes, which was attributed to better lactation persistency. 

The shift in the milking frequency of the flock from TAD to OAD in November strongly affected milk production. Ewes that lambed late missed the TAD milking period, producing 15.7, 1.0, 0.7, and 0.8 kg less milk, fat, protein, and lactose, respectively, compared to ewes that were milked for 14 days TAD in early lactation. Additionally, the lambing date itself can affect the milk production of ewes [67,68]. In the present study, the lambing date and days in TAD milking were correlated and were confounding effects if simultaneously included.

The strong effect of a TAD milking frequency in early lactation on fat yield affected the lactation persistency of fat. Ewes that were milked TAD in the first part of lactation produced a higher yield of fat in this period, and therefore, production declined more sharply after shifting to OAD, translating into worse lactation persistency compared to ewes that were milked OAD from the start. Other studies have confirmed a significant decline in milk production with the reduction from twice- to once-a-day milking [69,70].

## 5. Conclusions

There are a limited number of scientific publications on pasture-based dairy sheep production in New Zealand. The differences in milk production between the current study and other studies were not only attributed to the different genetic makeup of the animals but also due to the different environment, feeding, and lamb weaning systems. It is also important to note that due to the long exclusive lamb suckling period of the farm, this study only modelled the after-peak curves and not the full lactation curves. The low total yield produced is also a consequence of the sharp decrease in the lactation curves, which were largely influenced by the reduced pasture quality in this seasonal pasture-based system. This study provides experiences for a larger-scale animal evaluation and breeding programme to improve dairy sheep genetics for New Zealand farming systems.

## Figures and Tables

**Figure 1 animals-13-00349-f001:**
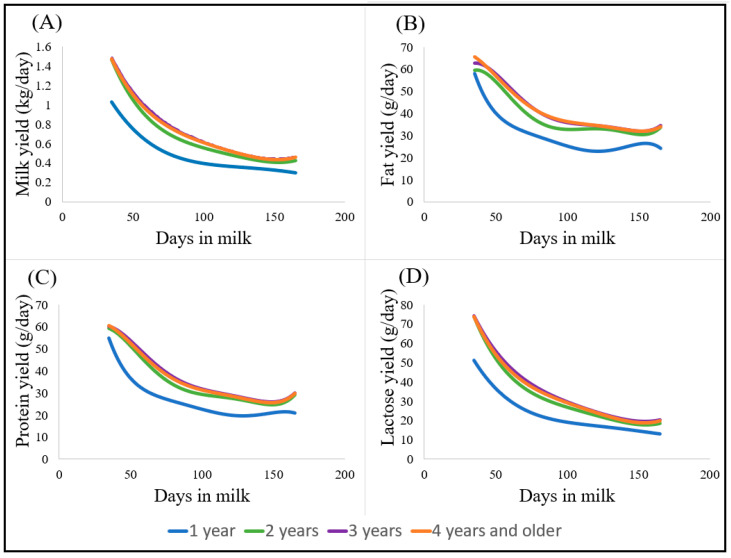
Lactation curves for daily yields of milk (**A**), fat (**B**), protein (**C**), and lactose (**D**) in lactations of Dairymeade ewes of one (blue), two (green), three (purple), and four years old or older (orange), modelled using orthogonal polynomials of order 4 for milk and lactose yields and of order 5 for fat and protein yields.

**Table 1 animals-13-00349-t001:** Pasture analyses using the near-infrared reflectance spectroscopy technique based on dry matter.

	DM	ME	CP	NDF	OMD
Month	(%)	MJ/kg DM	%DM	%DM	%
October	19.1	12.0	26.7	36.5	>84
November	19.9	11.4	17.3	45.9	77.2
December	24.7	10.5	13.5	49.2	72.5
January	29.7	8.2	14.1	60.6	57.6

DM = dry matter; ME = metabolisable energy; CP = crude protein; NDF = non-detergent fibre; OMD = organic matter digestibility.

**Table 2 animals-13-00349-t002:** Descriptive statistics of production variables considered in the genetic evaluation of Dairymeade sheep in 2021–2022 season (n = 169 lactations).

Trait	Mean	SD ^1^	Min	Max	CV ^2^ (%)
Lactation length ^3^ (days)	165	23.6	90	203	14
Milking length ^4^ (days)	108	25.8	35	133	24
Lactation yields ^5^ (kg)					
Milk	86.1	26.1	25.1	168.8	30
Fat	5.1	1.4	2.0	9.6	28
Protein	4.5	1.2	1.7	8.2	27
Lactose	4.1	1.3	1.2	8.0	31
Persistency ^6^ (%)					
Milk	55.8	8.9	35.9	94.9	16
Fat	72.7	11.1	53.5	115.5	15
Protein	65.7	9.5	46.2	99.2	15
Lactose	52.3	8.4	32.2	86.8	16
Average SCS ^7^	16.9	1.7	14.7	23.6	10
Age (years)	2.8	1.1	1	4	39
Litter size	1.7	0.6	1	3	33

^1^ SD = standard deviation. ^2^ CV = coefficient of variation. ^3^ Lactation length (from lambing to dry-off). ^4^ Milking length (from weaning to dry-off). ^5^ Lactation yields estimated from daily yields from 35 to 164 days of lactation. ^6^ Lactation persistency defined as yield from day 101 to 164 divided by yield from day 35 to 100, expressed as percentage. ^7^ Average of SCS, calculated as SCS = Log_2_(somatic cell count).

**Table 3 animals-13-00349-t003:** Measures of goodness of fit of the model of the lactation curves for milk and lactose using random regression with a fourth-order Legendre polynomial and for fat and protein using random regression with a fifth-order Legendre polynomial.

	Regression Line of A on P				
	*a*	*b*	RPE (%)	r	$\rho_{ccc}$
Milk (kg/day)	−0.03 ± 0.01	1.05 ± 0.01	11.65	0.975	0.969
Fat (g/day)	−4.23 ± 0.53	1.10 ± 0.01	12.67	0.962	0.952
Protein (g/day)	−2.65 ± 0.36	1.07 ± 0.01	9.92	0.977	0.972
Lactose (g/day)	−1.44 ± 0.37	1.04 ± 0.01	12.79	0.972	0.969

a and b are the intercept and slope of the regression line of the actual (A) on predicted (P) values; RPE = relative predicted error; r = correlation coefficient; $\rho_{\mathrm{ccc}}$ = concordance correlation coefficient.

**Table 4 animals-13-00349-t004:** F-values for effects of ewe coat colour, ewe age, litter size (LS), and days in twice-a-day (TAD) milking on milk production traits in Dairymeade sheep in the production season of 2021–2022.

Trait	Coat Colour	Age	LS	Days in TAD
Lactation yields ^1^ (kg)				
Milk	0.07	6.70 ***	0.63	19.86 ***
Fat	0.61	3.83 **	0.00	27.47 ***
Protein	0.01	4.38 **	1.12	20.51 ***
Lactose	0.03	6.71 ***	0.63	17.48 ***
Persistency ^2^ (%)				
Milk	1.90	3.55 *	1.96	0.00
Fat	1.29	1.55	1.69	10.96 ***
Protein	0.50	1.37	5.09 *	2.03
Lactose	1.33	3.34 *	1.46	0.45
Average SCC	0.06	0.51	0.09	0.01
Average SCS ^3^	0.94	1.70	1.48	0.79

^1^ Lactation yields estimated from daily yields from 35 to 164 days of lactation. ^2^ Lactation persistency defined as yield from day 101 to 164 divided by yield from day 35 to day 100, expressed as percentage. ^3^ Average of SCS, calculated as SCS = Log_2_(somatic cell count). Statistical significance is given as * *p* < 0.05; ** *p* < 0.01; *** *p* < 0.001.

**Table 5 animals-13-00349-t005:** Least-squares means (Mean) and standard errors (SE) of lactation yields of milk (kg), fat (kg), protein (kg), and lactose (kg) estimated for 130 days after weaning for different ewe ages (year), litter sizes, coat colours, and days in twice-a-day (TAD) milking at Kingsmeade farm during the production season 2021–2022.

		Milk		Fat		Protein		Lactose	
Effect	*n*	Mean	SE	Mean	SE	Mean	SE	Mean	SE
Age									
1	26	64.3 ^b^	5.4	4.1 ^b^	0.30	3.6 ^b^	0.30	3.1 ^b^	0.30
2	47	85.8 ^a^	3.6	5.0 ^a^	0.20	4.5 ^a^	0.17	4.2 ^a^	0.18
3	39	92.1 ^a^	3.6	5.2 ^a^	0.20	4.7 ^a^	0.17	4.4 ^a^	0.18
≥4	57	90.2 ^a^	3.0	5.3 ^a^	0.17	4.6 ^a^	0.14	4.3 ^a^	0.15
Litter size									
1	66	81.7	2.8	4.9	0.16	4.3	0.14	3.9	0.14
2	103	84.5	2.9	4.9	0.16	4.4	0.14	4.1	0.14
Coat colour									
Black	33	82.5	4.0	4.8	0.22	4.3	0.20	3.9	0.20
White	136	83.7	1.9	5.0	0.11	4.4	0.09	4.0	0.10
Days in TAD									
0	57	71.2 ^b^	3.2	4.1 ^b^	0.18	3.8 ^c^	0.15	3.4 ^b^	0.16
14	36	86.8 ^a^	4.0	5.1 ^a^	0.22	4.5 ^b^	0.19	4.2 ^a^	0.20
21	38	86.3 ^a^	4.0	5.3 ^a^	0.22	4.4 ^b^	0.19	4.2 ^a^	0.20
28	38	94.8 ^a^	4.2	5.6 ^a^	0.24	5.0 ^a^	0.20	4.5 ^a^	0.21

*n* = number of ewes within each category. ^a,b,c^ Least-squares means with different superscripts within effect are significantly different (*p* < 0.05).

**Table 6 animals-13-00349-t006:** Least-squares means (Mean) and standard errors (SE) of lactation persistency (%) ^1^ of milk, fat, protein, and lactose yields, for different ewe age, litter size, coat colour, and days in twice-a-day (TAD) milking at Kingsmeade farm during 2021–2022 season.

		Milk		Fat		Protein		Lactose	
Effect	*n*	Mean	SE	Mean	SE	Mean	SE	Mean	SE
Age									
1	26	61.5 ^a^	2.2	78.3	2.6	69.9	2.3	57.6 ^a^	2.1
2	47	53.7 ^b^	1.4	74.1	1.7	65.1	1.5	50.5 ^b^	1.4
3	39	54.5 ^b^	1.4	71.8	1.7	64.6	1.5	51.1 ^b^	1.4
>4	57	54.2 ^b^	1.2	72.8	1.4	65.0	1.3	50.9 ^b^	1.2
Litter size									
1	66	57.0	1.1	75.4	1.4	67.9 ^a^	1.2	53.4	1.1
2	103	55.0	1.2	73.1	1.3	64.4 ^b^	1.2	51.7	1.1
Coat colour									
black	33	54.8	1.6	75.4	1.9	65.5	1.7	51.5	1.5
white	136	57.2	0.8	73.1	0.9	66.9	0.8	53.5	0.7
Days in TAD									
0	57	55.7	1.2	78.6 ^a^	1.5	67.1	1.4	51.3	1.2
14	36	56.3	1.6	74.2 ^ac^	1.9	67.1	1.7	53.0	1.5
21	38	57.2	1.6	70.7 ^bc^	1.9	66.6	1.7	54.3	1.5
28	38	55.0	1.7	71.4 ^bc^	2.0	63.6	1.8	52.0	1.6

*n* = number of ewes within each category. ^1^ Lactation persistency (%) = (yield from day 101 to 164/yield from day 35 to 100) × 100. ^a,b,c^ Least-squares means with different superscripts within effect are significantly different (*p* < 0.05).

## Data Availability

The data are not publicly available due to privacy restrictions.

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
