# Peer review of "Modelling Lactation Curves for Dairy Sheep in a New Zealand Flock"

_animals, 2023, doi:10.3390/ani13030349_

Round 1

Reviewer 1 Report

I revised the manuscript on Modelling the lactation curve for dairy sheep in New Zealand flock. The topic addressed is relevant and aims to provide new data on ewe performances. The study is well designed, however, some results presentation and data interpretation need to be improved.

Major comments

The introduction should be improved. Indeed, the whole introduction seemed a justification of the methodological approach used. For instance, the paragraphs dedicated to why the Legendre polynomials model was used should be shorter. Some information could be provided on the new Zealand dairy sector, the amount of milk produced by cows, ewes, and tendencies.

Pasture quality was evaluated but not used to discuss the milk production results. As the Authors found differences in months in the quality of the pasture, it seemed really important to consider the effect of the season in this study. This may affect the milk production, quality, and lactation curve observed. Lines 303-305: the descending trend of the parameters is probably due to the reduced nutritional characteristics of the pasture. Even in the discussion (386-392), the authors lack to stress the effect of reducing pasture quality on the lactation curve. I suggest the authors implement a seasonal comparison of milk production and quality, rather than an analysis of the lactation curve, which here is influenced by the descending pasture quality.

Minor comments

Line 148-187. Milk persistency calculation should be described.

Line 223. I would suggest reorganizing the result section. As shown it seemed to just replicate the sub-titles in the methodology. For instance, descriptive statistics, measures of goodness of fit, and analysis of variance are not suitable result titles.

Lines 224-233. A standard presentation of these parameters is g/kg DM or %DM. The unit for OMD is %. Please check again.

Author Response

Review report form and additional remarks

Authors’ response

The introduction should be improved. Indeed, the whole introduction seemed a justification of the methodological approach used. For instance, the paragraphs dedicated to why the Legendre polynomials model was used should be shorter. Some information could be provided on the new Zealand dairy sector, the amount of milk produced by cows, ewes, and tendencies

We agree with this indication.

The introduction has now a short contextualization of the New Zealand dairy sheep industry.

The paragraphs about the use of Legendre polynomials are now shorter in the introduction.

Pasture quality was evaluated but not used to discuss the milk production results. As the Authors found differences in months in the quality of the pasture, it seemed really important to consider the effect of the season in this study. This may affect the milk production, quality, and lactation curve observed.

The whole flock is seasonal so we cannot evaluate the feed effect. However, we have presented the information of the feed and improved the discussion on the effect of feed on milk yield.

The effect of deviation from median lambing date (which is a time effect and related to what would be the “season effect”) is highly correlated to the effect of days in TAD, so they could not be used in the animal model simultaneously. This is explained in lines 204-206 (with no markup of track changes).

“The covariate days in TAD was used in the model instead of deviation from median lambing date as these traits were strongly correlated”

Deviation from median lambing date was also defined (line 116):

“A deviation from the median lambing date of the flock was calculated for each ewe (ewe lambing date – median lambing date of the flock).”

Lines 303-305: the descending trend of the parameters is probably due to the reduced nutritional characteristics of the pasture. 

Thank you for this observation.

This has now been emphasised in line 289.

“This descending trend is not only a physiological occurrence but it is also likely to be a result of reduced pasture quantity and quality in this seasonal pasture-based system.”

Even in the discussion (386-392), the authors lack to stress the effect of reducing pasture quality on the lactation curve. I suggest the authors implement a seasonal comparison of milk production and quality, rather than an analysis of the lactation curve, which here is influenced by the descending pasture quality.

As explained above, the whole flock is seasonal so we cannot evaluate the feed effect because it would be confounding effect between stage of lactation and feed.

Emphasis on pasture (feed) effect is given.

“In addition, the rapid decline in milk production was largely influenced by reduced pasture quality in this seasonal pasture-based system” (Line 307).

“The low daily production per ewe achieved in this extensive farming system can also be attributed to low forage availability in the summer caused by hot and dry weather conditions.” (Lines 324-326)

“Under hot or dry conditions, East Friesian ewes have been noted to produce low to moderate yields, and poor adaptability to Mediterranean climate and semi-arid conditions has been reported” (Lines 339-341).

“On pasture-based systems, animals that have low forage availability and low supplementation are not able to fully express their productive potential [45] and the peak may not be observed. The peak is also not observed in less selected animals [46]. In grazing systems, weather affects not only pasture availability but also comfort and stress in animals, therefore, also playing an important role in the shape of the lactation curves [47].” (lines 376-381)

Line 148-187. Milk persistency calculation should be described.

Milk persistency was previously defined in lines 175-179 of the methodology section.

Line 223. I would suggest reorganizing the result section. As shown it seemed to just replicate the sub-titles in the methodology. For instance, descriptive statistics, measures of goodness of fit, and analysis of variance are not suitable result titles.

The titles of the result section have now been changed to improve text readability and flow.

Lines 224-233. A standard presentation of these parameters is g/kg DM or %DM. The unit for OMD is %. Please check again

These have now been checked and corrected to %DM for CP and NDF, and to % for OMD.

Reviewer 2 Report

An interesting article about modelling lactation curves for dairy sheep.

The results are extended from a single flock to dairy sheep in New Zealand (see the Simple Summary). It is a bold conclusion. You may mention something like, this study provides experiences for a larger scale (national) breeding and evaluation.

Given the data, the authors need to conclude that after-peak curves are modeled. The full-lactation curves are not modeled!

Figure 1: Given Figure 2, I found this figure unnecessary.

The authors calculated the the concordance correlation coefficient. However, there is not much about it in the manuscript.

Please see below for minor comments.

L19: Rephrase "persistency of dairy sheep"

L24: "production season of 2021-2022". Please clarify here and elsewhere. There are many seasons (lambing, weaning, milking, etc). Please define the months. The seasons are of course different in different parts of the world. Clear presentation is needed.

L33: Not an interesting ending. Please finish with something like "the study on this single flock provides valuable experiences for a larger (e.g., national) breeding program".

L37-39: This is a very general statement. Fitting lactation curves is one of several ways.

L56: It is not clear what "variation" is referring at.

L57: Change "Ali and Schaeffer" to "Ali and Schaeffer [5]"

L59: Change this line to "and the performance of Wood [3], Wilmink [4], and Ali and Schaeffer [5] models were strongly affected"

L70: Rephrase "outputted"

L77: Change "calculation" to "the calculation"

L79: Change "analysis" to "the analysis"

L83: Change "(breed" to "(such as breed"

L84: Change "(age" to "(such as age"

L88: Change "lactose" to "lactose yield"

L96: "then only of East Friesian bloodlines". It is not clear what's happening here. You mean from here on only East Friesian bloodlines were mated to each other?!

L99: Change "11 hectares" to "11 hectares land"

L115: "622 test-day records". How many ewes?

L126: Change "lambing data" to "lambing date"

L162: No line break

L164: Add: "where n is the maximum polynomial order"

L173: "Toeplitz"?!

L180: Change "α5)" to "α5, depending on the trait)"

L186: I am not sure how this translates to lactation persistency! The higher the ratio, the lower the production after 100d (compared to 35-100). How did you come up with it? Any reference? How was the distribution of this ratio in your data?

L192: Here and elsewhere change "(SAS Institute Inc., Cary, NC, USA)" to a proper citation (i.e., include SAS in the list of references).

L196: Change "x" to "×"

L202-203: Change "and µP and µA are the means of predicted and actual values" to "µP and µA are the means, and P and A refer to predicted and actual values"

L208: Change "estimated" to "the estimated"

L215: No line break

Table 1: Why DM in percentage?

L235-245: A lot of it is repeating Table 2. Keep it short to couple of lines.

L254-256: I think this conclusion is based on the assumption that a = 0. Without this assumption, you may mention that the model tended to over-predict.

L265: Change "regression" to "the regression"

L267: Change "had most effect on most total yields" to "had the most effect on the total yields"

L270: Change "older groups of ewes" to "older ewes"

L280: "Coat colour". Any specific motivation for including this effect?

L290: Change "age" to "age (year)"

Table 5: Change ">4" to ">=4"

L330: Rephrase "artificially reared"

L331: "116 L". I wonder whether "L" is a standard abbreviation or you need to define it.

L341: Change "in summer" to "in the summer"

L342: "exclusive suckling period". I don't understand what "exclusive" means here.

L360: Change "fat" to "fat yield"

L364: Change "calculation" to "the calculation"

L366: Change "100 %" to "100%"

L369-373: Please delete these lines. SCC is completely out of context here.

L386-387: It is weird. Any decline comes after a peak, observed or unobserved. You may delete these lines.

L390: "The peak is also not observed in less selected animals". It is weird. What is the relationship between less selected animals and their peak not being observed? I suggest deleting this sentence.

L394: Change "were of" to "had a"

L399: Change this line to "to be more correlated with high lactation yield than with lactation persistency [50]"

L408-409: Change "with 3-year-old ewes producing the highest yields, and producing more" to "but 3-year-old ewes produced more"

L413: Change "One year old" to "One-year-old"

L414: Change this line to "yield, respectively, than 3-year-old ewes. Two-year-old ewes produced more than 4-year-old ewes. However,"

L416: Change "One year old ewes`" to "One-year-old ewes'"

L418-419: It is confusing. How can one calculate the Pearson correlation between two fixed effects?

L433: Change "twins had a trend to produce" to "twins produced"

Author Response

Review report form and additional remarks

Author’ response

The results are extended from a single flock to dairy sheep in New Zealand (see the Simple Summary). It is a bold conclusion. You may mention something like, this study provides experiences for a larger scale (national) breeding and evaluation

Thank you for this suggestion. A sentence has now been added to the conclusion.

Given the data, the authors need to conclude that after-peak curves are modeled. The full-lactation curves are not modeled!

Thank you for this suggestion, this part has been added to the conclusion.

Figure 1: Given Figure 2, I found this figure unnecessary

We agree with this indication, Figure 1 has now been removed.

The authors calculated the concordance correlation coefficient. However, there is not much about it in the manuscript.

A discussion of measures of goodness of fit was inserted in the discussion section.

L19: Rephrase "persistency of dairy sheep"

This has now been corrected to:

“…there is room for improvement of lactation yields and of lactation persistency in dairy sheep…”

L24: "production season of 2021-2022". Please clarify here and elsewhere. There are many seasons (lambing, weaning, milking, etc). Please define the months. The seasons are of course different in different parts of the world. Clear presentation is needed

This has now been corrected to:

“…records were obtained during the milk production season of 2021-2022 (from October to January).”

L33: Not an interesting ending. Please finish with something like "the study on this single flock provides valuable experiences for a larger (e.g., national) breeding program".

Thank you, this has now been corrected to:

“The study on this single flock provides valuable experience for a larger scale animal breeding programme in New Zealand.”

L37-39: This is a very general statement. Fitting lactation curves is one of several ways.

This has now been re-phrased to:

“Modelling the lactation curve is one of way of evaluating animals for genetic selection. Test-day records can be used to predict…”

L56: It is not clear what "variation" is referring at.

These sentences were removed.

L57: Change "Ali and Schaeffer" to "Ali and Schaeffer [5]"

These sentences were removed.

L59: Change this line to "and the performance of Wood [3], Wilmink [4], and Ali and Schaeffer [5] models were strongly affected"

These sentences were removed.

L70: Rephrase "outputted"

These sentences were removed.

L77: Change "calculation" to "the calculation"

This has now been corrected to:

“Several methods for the calculation of lactation persistency…”

L79: Change "analysis" to "the analysis"

This has now been corrected to:

“The total yield obtained for each animal can be used for the analysis of variance.”

L83: Change "(breed" to "(such as breed"

This has now been corrected to:

Several factors that may affect the milk production and the shape of the lactation curve in-clude genetics (such as breed and individual genotype),

L84: Change "(age" to "(such as age"

This has now been corrected to:

“…factors (such as age or parity…”

L88: Change "lactose" to "lactose yield"

This has now been corrected to:

“…total yields and persistency for milk, fat, protein, and lactose yield.”

L96: "then only of East Friesian bloodlines". It is not clear what's happening here. You mean from here on only East Friesian bloodlines were mated to each other?!

This has now been rephrased to:

“The breed was established in 1996, initially using Coopworth and Border Leicester dams and semen from European East Friesian sires. The resulting progeny was subsequently mated with only East Friesian sires.”

L99: Change "11 hectares" to "11 hectares land"

This has now been corrected to:

“The farm has 11 hectares land…”

L115: "622 test-day records". How many ewes?

This has now been corrected to:

“622 test-day records were gathered from 169 ewes between October 2021 and end of January 2022…”

L126: Change "lambing data" to "lambing date"

This has now been corrected to:

“A deviation from the median lambing date of the flock was calculated…”

L162: No line break

Removed line break (L185).

L164: Add: "where n is the maximum polynomial order"

This has now been corrected to:

“where ’s are the regression coefficients of the lactation curve of the population and ’s are random regression coefficients describing the lactation curve for animal i, n is the maximum polynomial order…”

L173: "Toeplitz"?

The name of the covariance matrix. Now it has been rephrased to:

“The best covariance structure of random residual was a diagonal-constant matrix (Toeplitz) for the modelling of repeated records on the same animal, also based on AIC and BIC values…”

L180: Change "α5)" to "α5, depending on the trait)"

Corrected as indicated.

“…(α0 to α4 or α0 to α5, depending on the trait)…”

L186: I am not sure how this translates to lactation persistency! The higher the ratio, the lower the production after 100d (compared to 35-100). How did you come up with it? Any reference? How was the distribution of this ratio in your data?

A sentence was inserted to indicate that “The higher this ratio, the higher is the persistency. A more persistent lactation will have a flatter curve, with the persistency proportion approaching one.”

L192: Here and elsewhere change "(SAS Institute Inc., Cary, NC, USA)" to a proper citation (i.e., include SAS in the list of references)

The proper citation and reference for SAS software has now been added.

L196: Change "x" to "×

Changed the symbol in the formula.

L202-203: Change "and µP and µA are the means of predicted and actual values" to "µP and µA are the means, and P and A refer to predicted and actual values"

Changed to "µP and µA are the means, and P and A refer to predicted and actual values".

L208: Change "estimated" to "the estimated

Changed "the estimated”.

L215: No line break

Removed line break.

Table 1: Why DM in percentage?

It is the percentage of DM in the feed, the rest is water.

L235-245: A lot of it is repeating Table 2. Keep it short to couple of lines

The excessive information that is repeated on the table has now been removed from the paragraph.

L254-256: I think this conclusion is based on the assumption that a = 0. Without this assumption, you may mention that the model tended to over-predict.

Thank you for raising this point. The estimates of the regression coefficient lines are presented in Table 3 to discuss better the goodness of fit the modelling of the lactation curves using the random regression models. Also discussed in lines 354-363.

L265: Change "regression" to "the regression"

Changed.

L267: Change "had most effect on most total yields" to "had the most effect on the total yields"

Changed.

L270: Change "older groups of ewes" to "older ewes"

Changed.

L280: "Coat colour". Any specific motivation for including this effect?

Coat colour could be an indicator of animal variety within this breed, and the performance of white and black ewes could be different due to different genetic make up. This was indicated in the description of the model.

L290: Change "age" to "age (year)

Changed.

Table 5: Change ">4" to ">=4"

Changed.

L330: Rephrase "artificially reared"

Changed to:

“Although higher or similar yields have been reported for other New Zealand flocks, those studies had a short lamb suckling period …”

L331: "116 L". I wonder whether "L" is a standard abbreviation or you need to define it

“L” is the international abbreviation for litres.

L341: Change "in summer" to "in the summer"

Changed.

L342: "exclusive suckling period". I don't understand what "exclusive" means here.

“Exclusive suckling” is a term used for the type of lamb weaning system.  It means that it is not a mixed regime of shared lamb suckling and machine milking. Different to dairy cow farming, where calves are removed from the mother after being born, most dairy sheep farms allow for the lambs to stay with the ewe and suckle milk during the first 3-4 weeks of life.

L360: Change "fat" to "fat yield"

Changed.

L364: Change "calculation" to "the calculation"

Changed.

L366: Change "100 %" to "100%"

Changed.

L369-373: Please delete these lines. SCC is completely out of context here.

Deleted paragraph.

L386-387: It is weird. Any decline comes after a peak, observed or unobserved. You may delete these lines.

Some lactation curves do not initiate with a peak (which is defined by a gradual increase followed by a decrease). The lactation curves with no peak are characterised by simply starting at a high point.

L390: "The peak is also not observed in less selected animals". It is weird. What is the relationship between less selected animals and their peak not being observed? I suggest deleting this sentence.

Animals that are not highly selected for milk production are the ones that do not produce high yields of milk. High yields of milk are associated with peak in the start of the lactation. Less selected animals have no peak, only a gradual decrease.

L394: Change "were of" to "had a"

Changed.

L399: Change this line to "to be more correlated with high lactation yield than with lactation persistency [50]"

Changed.

L408-409: Change "with 3-year-old ewes producing the highest yields, and producing more" to "but 3-year-old ewes produced more

Changed.

L413: Change "One year old" to "One-year-old

Changed.

L414: Change this line to "yield, respectively, than 3-year-old ewes. Two-year-old ewes produced more than 4-year-old ewes. However,"

Changed.

L416: Change "One year old ewes`" to "One-year-old ewes'"

Changed.

L418-419: It is confusing. How can one calculate the Pearson correlation between two fixed effects?

Pearson correlation was obtained between the variables age, litter size, days in TAD. This enables to get the big picture of the flock and understand tendencies.

L433: Change "twins had a trend to produce" to "twins produced"

Changed.

Round 2

Reviewer 1 Report

The manuscript was significantly improved and is now suitable for publication. Thank you for addressing my comments.